# GABA_A_ Receptor β3 Subunit Mutation N328D Heterozygous Knock-in Mice Have Lennox–Gastaut Syndrome

**DOI:** 10.3390/ijms24098458

**Published:** 2023-05-08

**Authors:** Gerald Ikemefuna Nwosu, Wangzhen Shen, Kirill Zavalin, Sarah Poliquin, Karishma Randhave, Carson Flamm, Marshall Biven, Katherine Langer, Jing-Qiong Kang

**Affiliations:** 1Department of Biochemistry, Cancer Biology, Neuroscience and Pharmacology, School of Graduate Studies, Meharry Medical College, Nashville, TN 37208, USA; 2Department of Neurology, Vanderbilt University Medical Center, Nashville, TN 37232, USA; 3Vanderbilt Brain Institute, Nashville, TN 37232, USA; 4Department of Biological Sciences, Vanderbilt University, Nashville, TN 37235, USA

**Keywords:** epilepsy, knock-in mouse model, *GABRB3*, developmental epileptic encephalopathy

## Abstract

Lennox–Gastaut Syndrome (LGS) is a developmental and epileptic encephalopathy (DEE) characterized by multiple seizure types, electroencephalogram (EEG) patterns, and cognitive decline. Its etiology has a prominent genetic component, including variants in *GABRB3* that encodes the GABA_A_ receptor (GABA_A_R) β_3_ subunit. LGS has an unknown pathophysiology, and few animal models are available for studying LGS. The objective of this study was to evaluate *Gabrb3*^+/N328D^ knock-in mice as a model for LGS. We generated a heterozygous knock-in mouse expressing *Gabrb3* (c.A982G, p.N238D), a de novo mutation identified in a patient with LGS. We investigated *Gabrb3*^+/N328D^ mice for features of LGS. In 2–4-month-old male and female C57BL/J6 wild-type and *Gabrb3*^+/N328D^ mice, we investigated seizure severity using video-monitored EEG, cognitive impairment using a suite of behavioral tests, and profiled GABA_A_R subunit expression by Western blot. *Gabrb3*^+/N328D^ mice showed spontaneous seizures and signs of cognitive impairment, including deficits in spatial learning, memory, and locomotion. Moreover, *Gabrb3*^+/N328D^ mice showed reduced β_3_ subunit expression in the cerebellum, hippocampus, and thalamus. This phenotype of epilepsy and neurological impairment resembles the LGS patient phenotype. We conclude that *Gabrb3*^+/N328D^ mice provide a good model for investigating the pathophysiology and therapeutic intervention of LGS and DEEs.

## 1. Introduction

Lennox–Gastaut Syndrome (LGS) is a severe early-onset epileptic encephalopathy with an unknown pathophysiology affecting 1–2% of all epilepsy patients and 1–10% of all children with epilepsy [1]. A triad of hallmark features characterize LGS: tonic, atonic, and atypical absence seizures; slow spike-wave (SSW) complexes or generalized paroxysmal fast activity on an electroencephalogram (EEG); and cognitive impairment, including developmental delay, intellectual disability, or autistic behaviors seen in 75–95% of LGS patients [1,2,3]. These typically begin between 1 and 7 years of age and progressively worsen. Due to severity and the drug-resistant nature of the seizures, individuals affected with LGS are 14 times more likely to experience early mortality than other epileptic patients [4,5]. The etiology of LGS is grouped under two umbrellas: unknown origin and identifiable. The latter includes structural abnormalities, metabolic pathologies, and a considerable genetic component [6]. 

To date, mutations in several genes have been characterized in patients with LGS, including *ALG13*, *SCN8A*, *STXBP1*, *DNM1*, *FOXG1*, *CHD2*, and *GABRB3* [7,8]. Notably, a variety of epilepsies, including LGS, are associated with genes encoding subunits of GABA_A_ receptors (GABA_A_R) [8,9,10,11], hetero-pentameric ligand-gated ionotropic receptors that mediate the majority of inhibitory neurotransmission in adult CNS. We find it particularly important to study mutants of genes encoding GABA_A_R subunits, since these tend to directly impair inhibitory neurotransmission, thus causing a perturbation in the inhibitory and excitatory balance that typically underlies epileptic seizures. Several mutations in *GABRB3*, which encodes the GABA_A_R β_3_ subunit, are implicated in LGS, childhood absence epilepsy, and infantile spasms, an infantile epilepsy that often progresses to LGS [12,13]. In particular, the GABRB3(*D120N*) and GABRB3(*N110D*) mutations have been identified in patients with LGS and infantile spasms, respectively. *Gabrb3^+/N110D^* and *Gabrb3^+/D120N^* knock-in mouse models exhibit many features of LGS and infantile spasms [11,14,15]. Pathophysiology in these models appears to stem from impaired GABA_A_R function, while GABA_A_R expression and localization are unaffected by the mutation. 

Recently, a rare de novo mutation (c. A982G p.N328D) was identified in a patient with LGS and the detailed molecular pathophysiology of the mutation has been characterized in vitro [10]. The N328 residue is found in the third transmembrane region conserved across GABA_A_ receptor subunits. Experiments in heterologous cells and rodent neuronal cultures show that β_3_ (N328D) expression decreases GABA-evoked currents, expression of GABA_A_Rs subunits, and surface and synaptic GABA_A_R subunit localization [10].

In this study, we investigated the contribution of *GABRB3*(N328D) to LGS pathology in vivo using a *Gabrb3^+/N328D^* knock-in mouse model that we generated. In particular, we characterized the hallmark features of LGS, including seizures, EEG activity, and neurobehavioral abnormalities. Additionally, we investigated pathologic changes in GABA_A_R subunit expression in the *Gabrb3^+/N328D^* mouse. We find that the *Gabrb3^+/N328D^* mouse serves as a relevant model for investigating receptor perturbation that underlies an LGS phenotype [11,15]. Moreover, our findings suggest that this epileptic syndrome has a novel pathophysiology in comparison to the two previously published models.

## 2. Results

### 2.1. Gabrb3^+/N328D^ Mice Displayed Seizure Types and Ictal EEG Patterns Consistent with LGS 

The mouse was generated with the CRISPR-CAS9 strategy to mediate A982G knock-in in the coding sequence (Figure 1A–C). Homozygous mice were rarely born and were not viable (Figure 1D). The primers specific for identifying the wild-type (WT) allele or the mutant allele were paired with a reverse primer common to both the wild-type and the mutant allele to distinguish the WT from the heterozygous mutant knock-in (KI) mice (Figure 1E,F). Heterozygous mice were bred with C57BL/6J wild-type mice (stock 000664). To test whether the *Gabrb3*^+/N328D^ mice recapitulate the seizure phenotype of LGS, we evaluated seizure activity in *Gabrb3*^+/N328D^ (N = 6) and WT littermates (N = 6) for 48 h using video electroencephalogram (vEEG) monitoring. *Gabrb3*^+/N328D^ mice spontaneously exhibited two types of seizures: atypical absence (Figure 2B,C) and tonic (Figure 2D–G). On the other hand, the WT littermates were seizure free as expected of the C57BL/6J background [16]. 

Atypical absence seizures are generalized seizures, marked by a staring spell consisting of blank and fixed expression along with motor arrest [17]. In comparison to “typical” absence seizures, atypical absence seizures have a slower onset, recovery, and a more indicative change in tone [18]. In LGS patients, this seizure type displays characteristic SSW discharges in hippocampal and thalamic EEG activity with a strong association with cognitive dysfunction [19,20,21,22]. Similarly to humans with LGS, *Gabrb3*^+/N328D^ mice showed atypical absence seizures, which were characterized by (1) an ictal EEG containing SSW discharges of high power and low frequency (5–7 Hz) and (2) behavioral arrest (Figure 2B,C,H,I) [23]. Analogously to LGS patient seizures, absence seizures in *Gabrb3*^+/N328D^ mice were determined to be atypical in comparison to absence epilepsy mouse models, where typical absence seizure duration is 0.3–10 s within a 6–8 Hz frequency range [24]. 

Tonic seizures in humans are characterized by either uni- or bilateral contraction of one or more muscle groups. The contraction is sustained, lasting for a few seconds, and may result in turning to one side [25]. Tonic seizures in humans show an ictal EEG demonstrating generalized poly-spikes, which can be obscured by muscle or movement artifacts [26]. *Gabrb3*^+/N328D^ mice showed tonic seizures involving a sudden and prolonged contraction of the limbs, with or without tail stiffening, that appeared as either no change in EEG amplitude or low-amplitude, high-frequency activity, as previously described [11,27]. Tonic seizures were typically accompanied by low-amplitude, high-frequency activity on the EEG (Figure 2D–G,J,K). However, as observed by several other studies, the EEG pattern was variable and served as a poor indicator of tonic seizures. Similarly to these studies, we used behavioral abnormalities as the primary source of tonic seizure identification [11,27]. 

### 2.2. Gabrb3^+/N328D^ Mice Had Impaired Learning Capabilities

Another hallmark feature of LGS is impaired cognitive abilities as observed in LGS patients and mutation-bearing *Gabrb3* knock-in mouse models with DEE and LGS [11,15]. It has since been hypothesized that epileptic processes associated with LGS, including the presence of SSWs, can lead to patterns of abnormal activity and connectivity, which may compete with normal development, resulting in cognitive impairment and regression [28,29]. Therefore, we tested if *Gabrb3*^+/N328D^ mice exhibit impaired spatial learning and memory using the Barnes maze (Figure 3). 

Both *Gabrb3^+/N328D^* (*N* = 9) and WT (*N* = 9) mice were tested. Learning ability was measured during an initial 5-day training period, during which time spent searching for the target hole and latency to first target hole entry were assessed. *Gabrb3*^+/N328D^ mice showed a significant learning deficit in comparison to WT littermates (Figure 3A–C). Memory capabilities were assessed through a 5-min probe trial occurring one hour after the final test session (Figure 3D–F), which also showed a significant decrease in maze search time and time spent searching for the target for *Gabrb3*^+/N328D^ mice in comparison to WT mice. The probe and training trials measured the same parameters to assess cognition. Next, the path efficiency was determined along with the search strategy used most per genotype. In comparison to WT, *Gabrb3*^+/N328D^ mice were significantly less efficient at completing the maze and used a random search strategy, indicating a significant learning deficit (Appendix A). 

### 2.3. Gabrb3^+/N328D^ Mice Did Not Display a Comorbid Autism Spectrum Disorder Phenotype

Autism spectrum disorder (ASD) shares a comorbidity in patients with LGS [3]. A core feature of ASD is impaired social interaction and communication, and anxiety is a common feature as well [30]. We tested the behavior of Gabrb3^+/N328D^ mice for these ASD features. We assessed anxiety behaviors using the elevated zero maze, which exploits the natural tendency of rodents to seek shelter for security, represented by closed arms of the maze [31]. Another marker of anxious behavior is the fear response displayed by mice as freezing episodes in the open arm. Compared to WT siblings (*N* = 8), Gabrb3^+/N328D^ mice (*N* = 8) showed no significant difference in the amount of time spent in the open vs. closed arms, nor in freezing behaviors (Figure 4A–C), suggesting that Gabrb3^+/N328D^ mice do not exhibit anxiety. However, Gabrb3^+/N328D^ mice showed a large locomotion deficit in comparison to WT littermates (Figure 4D). Next, we assessed social interaction of Gabrb3^+/N328D^ (*N* = 27) vs. WT (*N* = 27) mice using the three-chamber social neurobehavioral paradigm, which evaluates sociability of a test mouse with a “novel” mouse. Initial introduction of the “novel” mouse is followed by a period of exploration and interaction, during which the “novel” mouse becomes familiar. Subsequently, another new mouse is introduced, allowing for assessment of social preference for the new vs. familiar mouse. Both the sociability and social preference tasks showed non-significant deficits (Appendix A). 

To further evaluate the possible ASD phenotype, we monitored the home cage of *Gabrb3*^+/N328D^ (*N* = 4) and WT (*N* = 4) mice. This test has been used previously to monitor genetic models of ASD [32]. Home cage monitoring allows us to assess eating, drinking, sleeping, hang cuddle activity, grooming, and travelling behavior of mice across 72 h. We saw that *Gabrb3*^+/N328D^ mice did show a significant decrease in drinking and increase in sleeping in comparison to WT littermates (Appendix A). Eating was trending towards a deficit in *Gabrb3*^+/N328D^ mice as well, although it did not show significance. However, we did not observe ASD comorbidity in *Gabrb3*^+/N328D^ mice.

### 2.4. Gabrb3^+/N328D^ Mice Showed Diminished Expression of β3 Subunit in Cerebellar, Thalamic, and Hippocampal Tissue

GABA_A_Rs are hetero-pentamers primarily composed of α_1_β_2_γ_2_ (60%), α_2_β_3_γ_2_ (15–20%), and α_3_β_2,3_γ_2_ (10–15%) subunit combinations in the adult brain that differ in function and regional expression and often form with preferred partnering subunits [33]. We hypothesized that *Gabrb3*^+/N328D^ mice may show a pathologic change in expression of the β_3_ subunit as well as changes in expression of the α_2_ and α_3_ GABA_A_R subunits that partner with β_3_ [33] and compensatory changes in other major GABA_A_R subunits [34,35]. We quantified expression of α_1_, α_2_, α_3_, β_2_, β_3_, and γ_2_ subunits in the cerebellum, cortex, hippocampus, and thalamus by Western blot in WT and *Gabrb3*^+/N328D^ mice (*N* = 5 for α_1,_ β_3_, and γ_2_; *N* = 3 for α_2_, α_3_, and β_2_ for WT and *Gabrb3*^+/N328D^ mice). We saw a significant decrease in β_3_ in the cerebellum, hippocampus, and thalamus (Figure 5I,J). Moreover, we observed a trend towards decreased β_3_ expression in the cortex that was in line with decreased expression across the other regions, but this was not statistically significant. However, there was no significant change in the expression of α_1_, α_2_, α_3_, β_2_, β_3_, or γ_2_ subunits (Figure 5A–H,K,L).

## 3. Discussion

The goal of this study was to evaluate the *Gabrb3^+/N328D^* knock-in mouse as a suitable model of LGS and to investigate in vivo molecular pathophysiology of LGS associated with GABA_A_R mutations. We found that the *Gabrb3^+/N328D^* mouse recapitulates the major hallmarks of LGS, including major seizure types and EEG activity patterns characteristic of LGS patients as well as a deficit in locomotion and learning and memory capabilities, indicating cognitive decline. Additionally, we observed decreased drinking and increased sleeping behavior of *Gabrb3^+/N328D^* mice in home cage monitoring, with latter likely being associated with tonic seizures that disturb *Gabrb3^+/N328D^* mice during sleep. However, *Gabrb3^+/N328D^* did not display anxiety or deficient sociability that are characteristic of ASD, a comorbidity in a significant fraction of LGS patients. Parameters of home cage monitoring indicative of repetitive behaviors associated with ASD, such as obsessive self-grooming or hang cuddling, also did not show a difference. Finally, we showed reduced expression of the GABA_A_R β_3_ subunit in cerebellar, hippocampal, and thalamic regions that underlies the LGS pathology. Based on our results, the *Gabrb3^+/N328D^* mouse is a viable model that can be employed to further study LGS pathophysiology and therapeutic intervention.

### 3.1. Gabrb3^+/N328D^ Mice Exhibit Seizure Types Typical of LGS

*Gabrb3^+/N328D^* mice exhibited spontaneous tonic and atypical absence seizures, which are two primary seizure types observed in children affected with LGS. Tonic seizures are a hallmark feature of LGS, which are characterized by a sudden onset of stiffening of the body with or without a pattern of fast activity on the EEG [11,18], both of which we observed in *Gabrb3^+/N328D^* mice. Typically, tonic seizures have a focal onset, but we were unable to evaluate focal origin of seizures due to technical limitations. Similar tonic seizures have been observed in both *Gabrb3*^+/D120N^ and *Gabrb3*^+/N110D^ mouse models associated with LGS and infantile spasms, respectively [11]. However, in line with seizure types found in patients with LGS [36], tonic seizures are not the predominant seizure type in either model. Similarly to *Gabrb3*^+/D120N^ mice, *Gabrb3*^+/N328D^ mice experience atypical absence seizures significantly more frequently than tonic seizures. In fact, tonic seizures are also the least observed seizure type in *Gabrb3*^+/N110D^ mice.

Atypical absence seizures are another hallmark feature of LGS, which are characterized by sudden lapses in conscious awareness with corresponding spike-wave discharges on the EEG [37]. We observed both of these characteristics in *Gabrb3*^+/N328D^ mice. The ILAE has classified these type of seizures as generalized with a non-motor aspect [38], but due to aforementioned technical limitations, we could not evaluate the generalized vs. focal nature of the seizures. While *Gabrb3*^+/N328D^, *Gabrb3*^+/D120N^, and *Gabrb3*^+/N110D^ mice all exhibit atypical absence seizures, there is a spectrum of seizure severity between the models. *Gabrb3*^+/N328D^ mice had fewer atypical absence seizures than *Gabrb3*^+/D120N^ mice but more than *Gabrb3*^+/N110D^ mice [11,15]. Therefore, the D120N mutation appears to cause a more severe seizure phenotype. From a structural point of view, this is succinct, as the N328D is found in the third transmembrane region and therefore can have significant effect on protein stability as well as D120N, found in the binding loop, and N110D, with the least severe seizure phenotype, found on a non-functional region of the β3 subunit.

### 3.2. Gabrb3^+/N328D^ Mice Showed a Cognitive Decline in Learning Ability and Deficit in Locomotion

The *Gabrb3^+/N328D^* mouse displayed decreased locomotion in the elevated zero maze and learning and memory capabilities in the Barnes maze test for spatial learning and memory, which suggests a cognitive decline phenotype characteristic of developmental and epileptic encephalopathies and LGS in particular. The majority of LGS patients (75–95%) show cognitive impairment, including developmental delay, intellectual disability, or autistic behaviors [3]. Landau and Kleffner first hypothesized that not only the seizure activity but also interictal epileptiform activity can contribute to the decline in cerebral function despite stagnation of syndrome progression [39]. Cognitive impairment in epilepsy can affect memory, psychomotor function, and executive function deficits, depending on location of epileptic activity and whether it is focal or generalized [40]. In particular, learning and memory are often affected in LGS, for which reason we tested the *Gabrb3^+/N328D^* mice with the Barnes maze test for spatial learning and memory.

*Gabrb3^+/N328D^* showed a palpable decrease in spatial learning capabilities, as they took significantly longer than the WT mice to search for the target hole and make the first target hole entry. Similarly to *Gabrb3^+/N328D^* mice, both *Gabrb3*^+/D120N^ and *Gabrb3*^+/N110D^ mouse models show a deficit in learning capabilities [11]. When performing the Barnes maze test, all three models showed an increased latency to find the target hole in comparison to the WT littermates. Additionally, *Gabrb3^+/N328D^* mice showed a significant increase in time for finding the target hole during the probe trial, suggesting that there is a significant spatial memory deficit in *Gabrb3*^+/N328D^ mice. This finding also corroborates test results in *Gabrb3^+/D120N^* and *Gabrb3^+/N110D^* mice, which display a significant deficit in memory trials in comparison to their WT littermates.

The strategy that *Gabrb3^+/N328D^* mice used to find the target hole also significantly differed from WT mice and indicated a cognitive impairment. WT mice tend to use a spatial or serial approach to finding the target hole, whereas *Gabrb3^+/N328D^* mice show a random or serial mixed pattern across training days. A more random search strategy likely indicates that mutant mice are not able to identify visual cues across training days, suggesting a learning deficit. This observation is in line with *Gabrb3^+/D120N^* mice displaying a higher search strategy score than WT mice, which represents less efficient searching [11].

There is an important connection between seizure disorders and anxiety-like behavior [41]. Anxiety is a trigger for seizures in some patients, and we thus wanted to assess whether there was significant anxiety-like behavior in the *Gabrb3^+/N328D^* mouse model by using the elevated zero maze [31]. We did not observe anxious behavior in *Gabrb3^+/N328D^* mice based on time spent in open arms or freezing episodes. However, the *Gabrb3^+/N328D^* mice displayed a decreased level of locomotor activity compared to WT mice, suggesting a deficit in the ability to explore. This is in stark contrast to *Gabrb3^+/D120N^* mice, which travel the same distance in the elevated zero maze as WT mice. *Gabrb3^+/D120N^* mice also show hyperactivity in locomotor chambers, displaying no locomotor deficit. Moreover, *Gabrb3^+/D120N^* mice display impaired social interaction and anxiety, while *Gabrb3^+/N328D^* mice displayed no anxious behavior and normal sociability. This indicates a significant behavioral difference between the two models and suggests that *Gabrb3^+/N328D^* mice may not be as explorative as *Gabrb3^+/D120N^* mice. Indeed, *Gabrb3^+/N328D^* mice traveled a reduced distance only when exploring the elevated zero maze but showed no difference in distance travelled in their native environment during the home cage scan. Nonetheless, it is also possible that *Gabrb3^+/N328D^* mice suffer from a gait phenotype, as gait disturbances have been characteristic of LGS patients [42]. Therefore, investigating a pathology in gait of *Gabrb3^+/N328D^* mice is a viable future direction.

### 3.3. Gabrb3^+/N328D^ Mice Showed Reduced Expression of GABA_A_R β3 Subunit in Cerebellum, Hippocampus, and Thalamus

Brain regions, including the cerebellum, hippocampus, and thalamus, display reduced expression of the GABA_A_R β_3_ subunit in *Gabrb3^+/N328D^* mice. This suggests a generalized reduction of β_3_ subunit expression in the brain, as the cortex was the only region that did not show a statistically significant decrease but instead showed a trend towards reduced β_3_ expression. This finding corresponds with reduced expression levels of the β_3_ (N328D) subunit compared to the WT β_3_ subunit in cultured neurons, which suggests that the mutant protein is less stable. Moreover, β_3_ (N328D) showed significantly reduced synaptic localization in cultured neurons, and surface expression of β_3_ and other major GABA_A_R subunits was reduced in heterologous transfected cells when β_3_ (N328D) was co-transfected with other major WT subunits [10]. These in vitro experiments indicate that the β_3_ (N328D) subunit can affect stability and subcellular localization of GABA_A_Rs, including other partnering subunits. For this reason, we also measured expression of the major GABA_A_R subunits in *Gabrb3^+/N328D^* mice.

We hypothesized that reduced β_3_ subunit expression would be accompanied by reduced expression of α_2_ and to lesser extent α_3_ subunits, which have been found to preferentially form receptor complexes with β_3_, particularly in hippocampus, where expression of both α_2_ and β_3_ is typically high [43,44]. Furthermore, we hypothesized a mild compensatory upregulation of β_2_ and other major subunits, as has been seen when expression of other subunits is lost [45]. However, we did not find a difference in total expression of major GABA_A_R subunits other than the β_3_ subunit in any of the tested brain regions. It is possible that while total expression of these subunits appears unchanged in *Gabrb3^+/N328D^* mice, their subcellular localization to the synapse and membrane is altered as in in vitro models [10]. These changes in subcellular localization could be detected by measuring surface expression or determining co-localization of subunits with synaptic markers, which is a viable direction for future investigation.

Expression deficits seen in *Gabrb3^+/N328D^* mice here and with the recombinant receptor β_3_(N328D) subunit expression in vitro by previous studies [10] are not observed with *Gabrb3^+/D120N^* and *Gabrb3^+/N110D^* mice. This indicates that β_3_(N328D) expression is different than with these models. For instance, total expression and surface trafficking of α_1_, β_3_, and γ_2_ subunits are comparable to WT in *Gabrb3^+/D120N^* mouse brain tissue [10]. Similar expression data are not yet available for *Gabrb3^+/N110D^* mice, but in vitro experiments in heterologous cells show no decrease in total nor surface expression of major GABA_A_R subunits, including the β_3_ subunit, when the β_3_(N110D) or β_3_(D120N) subunit cDNAs are co-expressed with other subunits [46]. While it is common for GABA_A_R epilepsy mutations to produce receptors with altered expression, these results suggest that β_3_(N110D) and β_3_(D120N) mutant subunits can compete with wild-type subunits to form a functionally impaired receptor. It appears that the pathophysiology of *Gabrb3^+/D120N^* and *Gabrb3^+/N110D^* involves impaired GABA_A_R channel function, while impaired expression and/or trafficking is a significant contribution to *Gabrb3^+/N328D^* pathology [11,47]. However, it is likely that in the end, both reduced expression in this mouse model and function impairment in the other two models converge on diminishing inhibition in vital epileptogenic circuits, such as the thalamocortical circuit, as shown for *Gabrb3^+/D120N^* and *Gabrb3^+/N110D^* mice [11,15]. Evaluating similar changes in GABA_A_R-mediated synaptic currents in *Gabrb3^+/N328D^* mice, especially in light of diminished β_3_ expression that we have shown here, is another viable direction of future investigation.

An outstanding question remains as to how regional disruption contributes to the pathophysiology of seizures and cognitive decline of *Gabrb3^+/N328D^* mice. Region-specific loss of the β_3_ subunit suggests that the hippocampus, thalamus, and cerebellum are most affected by the mutation, and GABA_A_R-mediated inhibition in these regions may be deficient. For instance, the hippocampus is commonly known for its role in memory formation and retrieval, and it has been shown to have a high expression of β_3_ in the adult brain [48,49]. Therefore, deficiencies in GABAergic inhibition within the hippocampus could underlie deficits in learning ability of *Gabrb3^+/N328D^* mice. On the other hand, pathologies in the thalamocortical circuit are primarily responsible for generating absence seizures. A decrease in thalamic β_3_ subunit expression in *Gabrb3^+/N328D^* mice may contribute to decreased regional inhibition that underlies the pathophysiology of atypical absence seizures. However, this hypothesis requires electrophysiological testing of the multiple components of thalamocortical circuit. GABAergic inhibition plays multiple roles in this circuit, and both increases and decreases in regional inhibition can cause epileptogenic synchronization of thalamocortical oscillations [50]. For instance, a recent study demonstrated that both gain-of-function and loss-of-function variants in *GABRB3* are associated with epilepsy syndromes [51]. In addition, both seizures and cognitive decline can be caused by focal seizures that spread to other circuits; for instance, a seizure with a hippocampal focus can spread to the thalamocortical circuit. Furthermore, circuit development of *Gabrb3^+/N328D^* mice could be affected by pathological development, as β_3_ is expressed early during development and plays an important role in brain formation [52]. Therefore, potential dysfunction of thalamic or hippocampal inhibition and its effects on thalamocortical oscillations remain to be investigated through electrophysiological means in *Gabrb3^+/N328D^* mice. Finally, diminished GABAergic inhibition in the cerebellum may contribute to motor and gait deficits that we discussed earlier, though the locomotor phenotype we observed appeared to be mainly behavioral. A separate study focusing on mouse gait and locomotion of *Gabrb3^+/N328D^* mice is needed to investigate this direction.

### 3.4. Gabrb3^+/N328D^ Mice Are a Viable Model for Studying Lennox–Gastaut Syndrome Pathophysiology and Therapeutic Intervention

Much is still unknown about the pathophysiology of LGS, and accurate animal models for investigating this disorder and testing potential therapies are much needed in the field. The *Gabrb3^+/N328D^* mouse can serve as a relevant model in this regard, as heterozygous expression of this LGS patient variant in mice reliably recapitulates hallmark features of LGS in humans. One particular opportunity granted by this model is to study potential therapy using in vivo overexpression of ubiquilin-1 (Plic-1), a protein that connects GABA_A_Rs to the ubiquitin protease system. Research has shown that in the instance of epileptic phenotypes, seizure activity reduces expression of Plic-1, and its overexpression can reduce the seizure phenotype, likely by helping clear misfolded mutant proteins from the trafficking pathway [53]. We hypothesize that overexpression of this protein can rescue the developmental and epileptic phenotype through exploiting both the chaperoning and recycling capabilities of Plic-1, and this is an exciting direction of therapeutic development for LGS that employs the *Gabrb3^+/N328D^* mouse in our future study.

## 4. Materials and Methods

### 4.1. Subjects

The mutant knock-in mice were bred with the C57BL/6J wild-type mouse (stock 000664). Both male and female C57BL6/J mice aged 2–4 months old were used in the study. Animals were housed in single-sex cages in maximum groups of five in a light- and temperature-controlled environment on a 12 h light/12 h dark schedule with free access to food and water. Male and female WT (control) and *Gabrb3^+/N328D^* heterozygous (experimental) mice of 2–4 months of age, corresponding to adolescence in mice, were used for the study. Sample size was calculated by power analysis using the G*Power software version 3.1.9.6. All procedures were approved by the Vanderbilt University Institutional Animal Care and Use Committee and were conducted in accordance with the NIH guide for the Care and Use of Laboratory Animals. We excluded from the study animals that showed any signs of distress prior to testing, including decreased activity, ungroomed appearance, abnormal stance, or excessive licking and scratching.

### 4.2. Generation of the Gabrb3^+/N328D^ Knock-in Mouse

*Gabrb3^+/N328D^* was constructed in collaboration with the Center for Mouse Genome Modification at the University of Connecticut Health center. The *Gabrb3^+/N328D^* mouse line was generated with a targeted A982G mutation at exon 8 of the endogenous *Gabrb3* locus. The procedures for the knock-in mouse construction are the same as previously described for *Gabrb3^+/D120N^* [11,15]. Each mouse was genotyped using two primer sets (G3-N328Df 5′-CTTCTGGAGTACGCCTTTGACG+G3-I8R 5′-GACCACTCCTGATCCTCCTCTG) to identify the mutant allele and (5′-CTTCTGGAGTACGCCTTTGTCA+G3-I8R 5′-GACCACTCCTGATCCTCCTCTG) to identify the wild-type allele. An annealing temperature of 62 °C was used for PCR reactions. Both bands appeared at 312 bp, so each mouse was genotyped with each primer set to identify the presence of the knock-in-specific allele.

### 4.3. Video-EEG Recordings and Analysis

Video-electroencephalograms were recorded for *Gabrb3^+/N328D^* and WT littermate mice. EEG electrodes were implanted following a previously published protocol [54]. Briefly, mice were anesthetized using 2–5% isoflurane and placed into a stereotaxic unit. They were then administered 5 mg/kg of ketoprofen via subcutaneous injection. The head mount was placed on the skull with the front edge 3.0 to 3.5 mm anterior of the bregma and secured with four screws (#820: Pinnacle Technology) resting in the cerebral cortex, providing electrical contact between the brain surface and the head mount [54,55,56]. Two EMG leads were bilaterally inserted above both trapezius muscles. After a 5–7-day recovery period, video-EEG monitoring was begun for 48 h with mice freely moving in the chamber. The head mounts were connected to a pre-amplifier (#8202-SE), which connected to the three-channel data acquisition and conditioning system (#8206-SE, Model 4100) to record from two EEG channels and one EMG channel. The acquisition rate for these channels was set to 400 Hz. The pre-amplifier filtered at 1 Hz high pass and amplified the signal at a gain of 100×. Video-EEG-EMG data were analyzed offline with Sirenia^®^ Seizure software version 9037-K through a power analysis. A baseline power of 1000 uV^2^ was used for seizure identification. Seizures were scored by a blinded and skilled scorer. Recordings were analyzed in their entirety for number of seizures, seizure duration, type, and interictal period. The Racine scale was used for grading of seizure severity based on behavior [57]. The following stages defined seizure severity: (0) no abnormality; (1) mouth and facial movements; (2) head nodding; (3) forelimb clonus; (4) rearing; and (5) rearing and falling [57].

### 4.4. Mouse Behavioral Testing

The mouse behavior experiments were performed using the Murine Neurobehavior Core lab at the Vanderbilt University Medical Center. Each cohort of *Gabrb3^+/N328D^* and WT littermate mice underwent a three-chamber socialization test, Barnes maze test, and elevated zero maze, with procedures following published protocols [58,59,60]. If mice showed obvious seizure activity, testing was terminated and suspended for the day to allow recovery.

#### 4.4.1. Three-Chamber Social Interaction Test

For the three-chamber socialization test, mice were allowed to acclimate to the testing room for 30 min. A habituation period was begun, which involved placing the subject mouse in the center chamber with the two gates down, thus initially separating the mouse from the other two chambers. The gates were lifted, and the subject mouse was allowed to explore all three empty chambers for 10 min to acclimate to the testing chamber. The subject mouse was then returned to the center chamber, and the gates were closed for preparation of the sociability stage. In the sociability stage, a stimulus mouse (novel mouse) was placed inside a wire pencil cup in one side chamber with an empty pencil cup (novel object) introduced in the opposite side chamber. The gates were opened to allow the subject mouse to explore all three chambers for 10 min. After 10 min, the subject mouse was returned to the middle chamber, and the gates were closed in preparation for the social novelty task. For the social novelty task, the empty pencil cup was replaced by a second stimulus mouse (novel mouse), with the original stimulus mouse now becoming a familiar social stimulus (familiar mouse). The gates were opened, and the subject mouse was allowed to explore all three chambers for 10 min. The time that the subject mouse spent within 1 cm of each stimulus for each of the stages was recorded using the ANY-maze software [61]. The maze was cleaned with 10% ethanol between subject mice to minimize odor cues.

#### 4.4.2. Barnes Maze

The Barnes maze test was broken up into three testing stages: habituation, training, and memory probe. For habituation, the mouse was placed in the start box for 30 s and then guided to the target hole, where the mouse was allowed to climb down into the escape box. After 30 s, the mouse was removed from the escape box and returned to the start box for three additional trials. All habituation trials were conducted for each mouse prior to training sessions. This session only occurred for the first day to acquaint the mice with the task and for the mice to realize that escape is possible. The training period commenced immediately after the habituation trials. Each trial began with the start box positions in the center of the maze, and the mouse was placed inside it. The mouse remained in the start box for 30 s with the lid closed, providing a standard starting context for each trial and ensuring that the initial orientation of the mouse on the maze varied randomly between trials. After 30 s in the start box, the box was lifted, and recording began. On each training trial, 11 of the 12 holes were blocked, and the remaining hole provided access to the escape box. Each mouse was permitted to explore the maze freely. After the mouse entered the escape box, the hole was covered to prevent the mouse from resurfacing onto the maze. The mouse was left in the escape box for 30 s before being returned to its home cage. If the mouse did not enter the escape box within 300 s, it was gently picked up by the base of the tail, placed in the palm of the hand, and let down at the side of the escape hole. The hole was then covered for 30 s before the mouse was returned to its home cage to prevent re-entry onto the maze. The maze was rotated clockwise every three holes between trials to prevent the use of intra-maze cues. Mice underwent four training trials per day for five consecutive days with an inter-trial interval of approximately 20 min. The first trial of each mouse was carried out before beginning the second trial. A 300 s probe test was conducted 1 h after the final spatial learning trial on the fifth day. This test was identical to the previous training trials, except that all 12 holes were blocked, and the maze was turned 180 degrees from the target hole of the last learning trial. Path efficiency, time spent searching for the target hole (defined as time getting closer or further from the target), and latency to find the target hole were recorded using ANY-maze software [61]. Search strategies were manually scored and defined as either a spatial path, serial path, or random path.

#### 4.4.3. Elevated Zero Maze

For the elevated zero maze, *Gabrb3^+/N328D^* mice were placed in an open arm to start. The maze itself was designed as a circle with two areas with walls that protrude upwards, thereby creating an open and closed area to the maze. The mice were then allowed to explore the maze for 5 min. Sheltering behavior (time spent in the closed vs. open arm), freezing behavior, and total distance traveled per genotype were recorded and analyzed using ANY-maze software [61].

#### 4.4.4. Home Cage Monitoring

Home cage monitoring took place in the Vanderbilt University Mouse Neurobehavioral Core. Mice were videotaped in their home cages for up to 72 continuous hours (12 h during the dark cycle and 6 h during the light cycle). The tapes were then coded by a trained observer. Proximity of mice as well as frequency and type (e.g., sniffing, grooming, tail-pulling, mounting, etc.) of social interaction were recorded. Number and duration of sleep bouts were noted as well as proximity of mice while sleeping. The amounts of food and water consumed in the home cage were measured on a daily basis.

### 4.5. Western Blotting

The cerebellum (CB), cortex (CTX), hippocampus (HP), and thalamus (TH) were dissected from 2–4-month-old *Gabrb3^+/N328D^* and WT mouse brains and homogenized with radio immunoprecipitation assay buffer (RIPA). Proteins were fractionated by 11% sodium dodecyl sulfate-polyacrylamide gel electrophoresis (SDS-PAGE) and immunoblotted with antibodies against the following antigens: anti-GABA_A_R α_1_ subunit (Millipore, Lot: 3202225, 1:1000 dilution, rabbit pAb), anti-GABA_A_R α_2_ subunit (Novus, Lot: WS211G, 1:1000 dilution, rabbit pAb), anti-GABA_A_R α_3_ subunit (Abcam, ab72446, 1:1000 dilution, rabbit pAb), anti-GABA_A_R β_2_ subunit (Millipore, Lot: 3491807, 1:1000, rabbit pAb), anti-GABA_A_R β_3_ subunit (Novus, Lot: KS417a, 1:1000 dilution, rabbit pAb), anti-GABA_A_R γ_2_ subunit (ABClonal, Lot: 1000090101, 1:1000 dilution, rabbit pAb), anti-glyceraldehyde 3 phosphate dehydrogenase (Abcam, ab8245, 1:1000 dilution, mouse mAb), recombinant anti-sodium-potassium ATPase (Abcam, ab76020, 1:1000 dilution, rabbit mAb). Membranes were then washed with 1x phosphate-buffered saline with tween-20 and incubated with IRDye^®^ 680RD Goat anti-Mouse IgG, IRDye^®^ 800CW Goat anti-Rabbit IgG secondary antibodies at a 1:5000 dilution. Immunofluorescence was detected via an Odyssey scanner. Band intensity was quantified using ImageJ densitometry software version 1.0, and each sample was normalized using glyceraldehyde 3 phosphate or ATPase as a loading control. Subsequently, the expression of the protein in the mutant animal was normalized to WT. Relative integrated density values were quantified and graphed using GraphPad Prism software version 9.2.0.

### 4.6. Statistical Analysis

We used GraphPad Prism software version 9.2.0 to perform statistical analysis and generate graphs. Tests were performed on each data set and are outlined in the corresponding figure legend. Data were expressed as mean ± standard error of the mean (SEM). For the majority of data sets, we used one- or two-way analysis of variance with Sidak’s multiple comparisons post hoc test to evaluate individual mean comparisons where appropriate. All analyses used an alpha threshold of 0.05 to determine statistical significance, and significance thresholds of 0.05, 0.01, 0.001, and 0.0001 were used for *, **, ***, and **** significance, respectively. In some cases, an unpaired *t*-test or one sample t-test was performed, and statistical significance was taken as *p*  <  0.05.

## Figures and Tables

**Figure 1 ijms-24-08458-f001:**
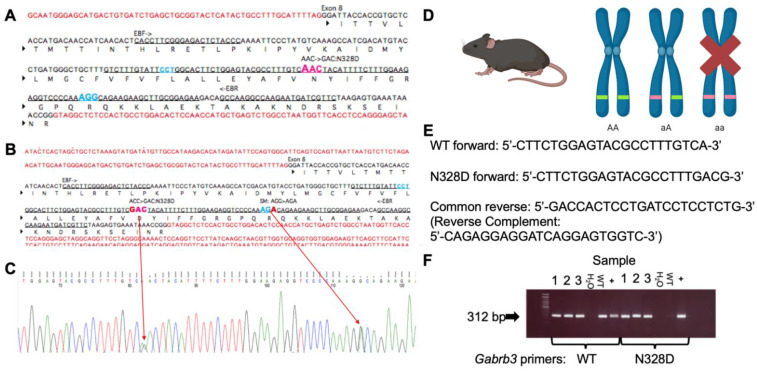
Generation of the *Gabrb3*^+/N328D^ knock-in mouse. (**A**) *Gabrb3* genomic sequence. The intended mutation is AAC > GAC on exon 8. AGG is the PAM site, and an AGG > AGA silent mutation was inserted to prevent guide cleavage. (**B**,**C**) Genomic sequence and representative chromatogram of the F1 generation showing insertion of intended mutation resulting in the N328D missense mutation. (**D**) Mice produced are genotypically wild-type (WT, AA) or heterozygous mutant (aA). Homozygous mutant mice (aa) are not viable and are often stillborn. (**E**) Genotype was confirmed by PCR. The forward mutant (KI), forward WT, and common reverse primers were used for identification of genotype in mice. Sequence complementary to the reverse primer is shown in parentheses. (**F**) WT and mutant bands both run at 312 bases with specific primer sets for WT and the mutant KI mice. 1, 2, and 3 correspond to heterozygous mutant samples, WT is wild-type tissue, and (+) represents heterozygous tissue.

**Figure 2 ijms-24-08458-f002:**
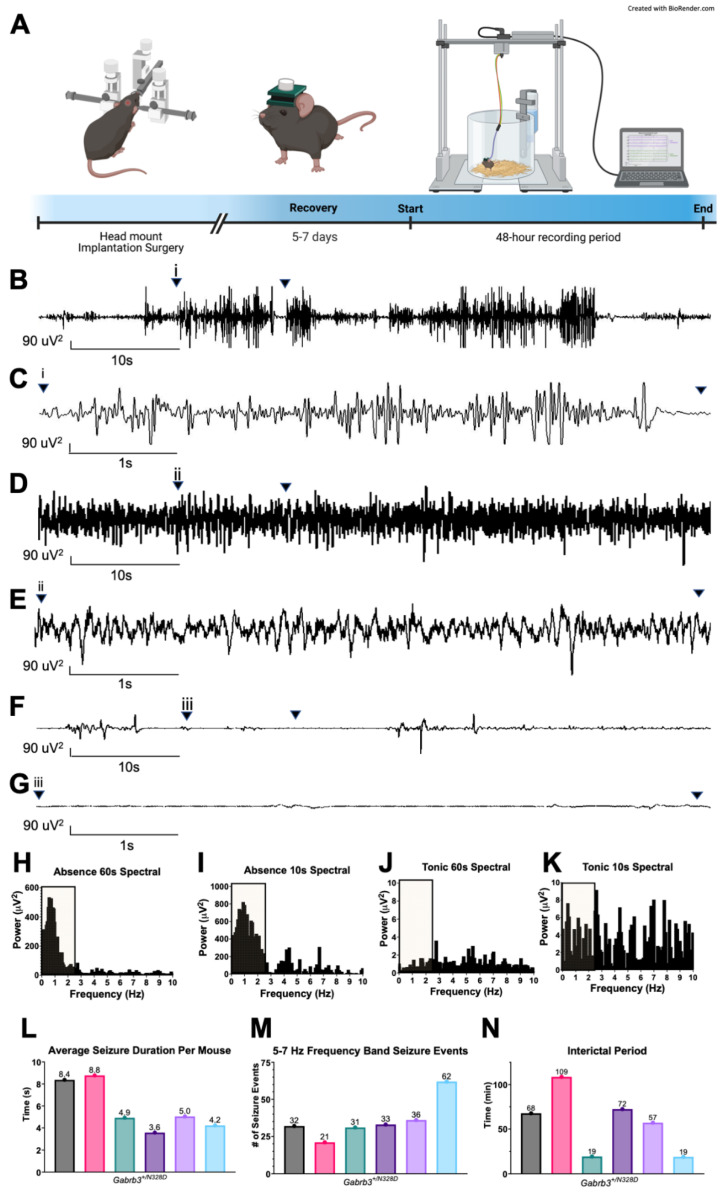
*Gabrb3^+/N328D^* mice show spontaneous seizures, SSW discharges, and high power in low frequency on the EEG. (**A**) Schematic diagram depicting the EEG head mount implantation and recording. (**B**) 1 min time span representative EEG trace of 2-month-old *Gabrb3^+/N328D^* mice had atypical absence seizure-like activity. (i) represents the start point for seizure activity and the 10 s time span seizure trace. (**C**) 10 s time span segment of (B). (**D**) 1 min time span representative EEG trace from 2-month-old *Gabrb3^+/N328D^* mice showing tonic seizure activity. (**E**) 10 s time span segment of (**D**). (**F**) 1 min time span EMG corresponding to (**D**) 1 min time span EEG trace. (**G**) 10 s time span segment of (**F**). (i, ii, and iii indicate the different seizure types.) (**H**–**K**) Spectral plots showing power level for 1 min and 10 s seizure segments from absence and tonic seizures, respectively. (**L**) Seizure duration ranged from 4 to 9 s with an average of 5.82 s. (**M**) Average occurrence of seizure activity irrespective of type was determined over a 48 h recording period alongside seizure duration and interictal period. *Gabrb3^+/N328D^* mice showed on average 36 seizures over a 48 h recording period. (**N**) The interictal period was on average 57 min for *Gabrb3^+/N328D^* mice (*N* = 6 WT littermates, *N* = 6 *Gabrb3^+/N328D^* mice) (Colored bars indicate separate mice).

**Figure 3 ijms-24-08458-f003:**
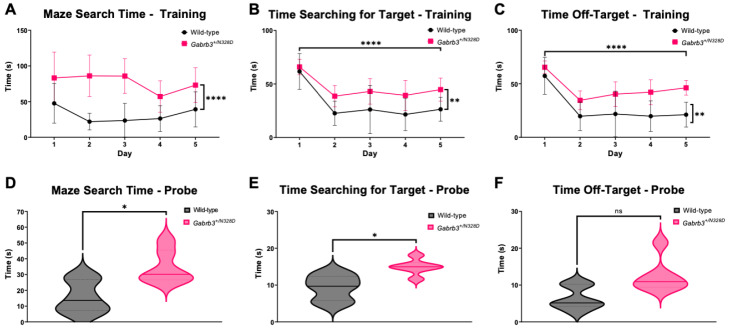
*Gabrb3^+/N328D^* mice show a significant learning and memory deficit on Barnes maze test. (**A**–**C**) *Gabrb3^+/N328D^* mice spend more time searching for the target hole assessed by time getting further from or closer to the target hole and latency to first entry to target hole. (**A**) Mouse genotype shows a very significant effect across training days indicated as each mouse spending more time getting further from the target hole. (**B**) Time getting closer to the target hole works the opposite in that less time should be spent getting closer to the target hole, indicating acknowledgment of target hole area. Genotype affected the results very significantly, with WT mice taking less time to get closer to the target area. (**C**) Genotype had an extremely significant effect on latency to target hole first entry. This shows that *Gabrb3^+/N328D^* mice have an extremely significant learning deficit as displayed by a longer latency to target hole entry across days in comparison to WT mice. ([WT: Day 1: 57.400 ± 7.041; Day 2: 19.708 ± 5.441; Day 3: 21.808 ± 8.681; Day 4: 19.742 ± 5.830; Day 5: 21.125 ± 4.725] [Het: Day 1: 65.450 ± 2.395; Day 2: 34.567 ± 3.562; Day 3: 40.325 ± 4.713; Day 4: 42.183 ± 4.725; Day 5: 46.275 ± 2.806]), time getting closer to the target hole ([WT: Day 1: 61.700 ± 6.822; Day 2: 22.642 ± 4.588; Day 3: 26.133 ± 9.178; Day 4: 21.542 ± 6.137; Day 5: 26.350 ± 4.541] [Het: Day 1:65.917 ± 2.949; Day 2: 38.617 ± 4.039; Day 3: 43.067 ± 4.820; Day 4: 39.375 ± 5.722; Day 5: 44.767 ± 4.413]), and latency to first target hole entry ([WT: Day 1: 47.550 ± 11.347; Day 2: 21.917 ± 4.746; Day 3: 23.467 ± 9.801; Day 4: 26.267 ± 7.364; Day 5: 39.192 ± 10.039] [Het: Day 1: 59.167 ± 19.856; Day 2: 68.233 ± 13.337; Day 3: 85.808 ± 9.914; Day 4: 50.925 ± 11.673; Day 5: 58.492 ± 13.321]) (**** *p* = 0.0001, and * *p* = 0.026, respectively). (**D**–**F**) Probe trials showed a significant difference in memory capabilities in *Gabrb3^+/N328D^* mice in comparison to WT littermates. Latency to first target hole entry indicated as maze search time ([WT: 15.83 ± 4.195] [Het: 34.80 ± 4.297] * *p* = 0.0102), time getting closer to the target hole indicated as time searching for target ([WT: 9.30 ± 1.4] [Het: 14.92 ± 0.8522] * *p* = 0.0246), and time getting further from target hole indicated as time off-target ([WT: 6.4 ± 1.286] [Het: 12.47 ± 1.889] *p* = 0.0717) (N = 6 per genotype). * *p* < 0.05, ** *p* < 0.01, and **** *p* < 0.0001 by two-way ANOVA with post hoc Sidak’s multiple comparison’s test for (**A**–**C**) and Student’s unpaired *t*-test for (**D**–**F**). ns = not significant.

**Figure 4 ijms-24-08458-f004:**
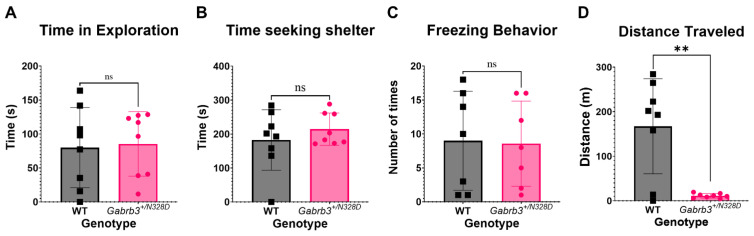
*Gabrb3^+/N328D^* mice display a pronounced locomotion deficit and no anxiety phenotype. *Gabrb3^+/N328D^* and wild-type mice were assessed via elevated zero maze for an anxiety phenotype. (**A**,**B**) Time spent in either the closed or open arm was assessed, and no significant difference was seen in mice (open: WT (79.88 ± 20.89), *Gabrb3^+/N328D^* (85.45 ± 16.78), *p* = 0.8382; closed: WT (182.6 ± 31.41), *Gabrb3^+/N328D^* (214.6 ± 16.78)). (**C**) Freezing episodes in the open arm were utilized as a marker for anxiety-like behavior as well (WT (1 ± 0.3651), *Gabrb3^+/N328D^* (0.6250 ± 0.3239), *p* = 0.4588, N = 8). There was a comparable number of episodes between genotypes, suggesting there is no significant anxiety-like behavior in the rodents. (**D**) However, the distance that *Gabrb3^+/N328D^* mice traveled over the testing period was significantly reduced in comparison to wild-type littermates (WT (167.3 ± 37.74), Het (11.27 ± 1.774), ** *p* = 0.0010, N = 8 for each genotype)., ** *p* < 0.01, and with Student’s unpaired *t*-test. ns = not significant.

**Figure 5 ijms-24-08458-f005:**
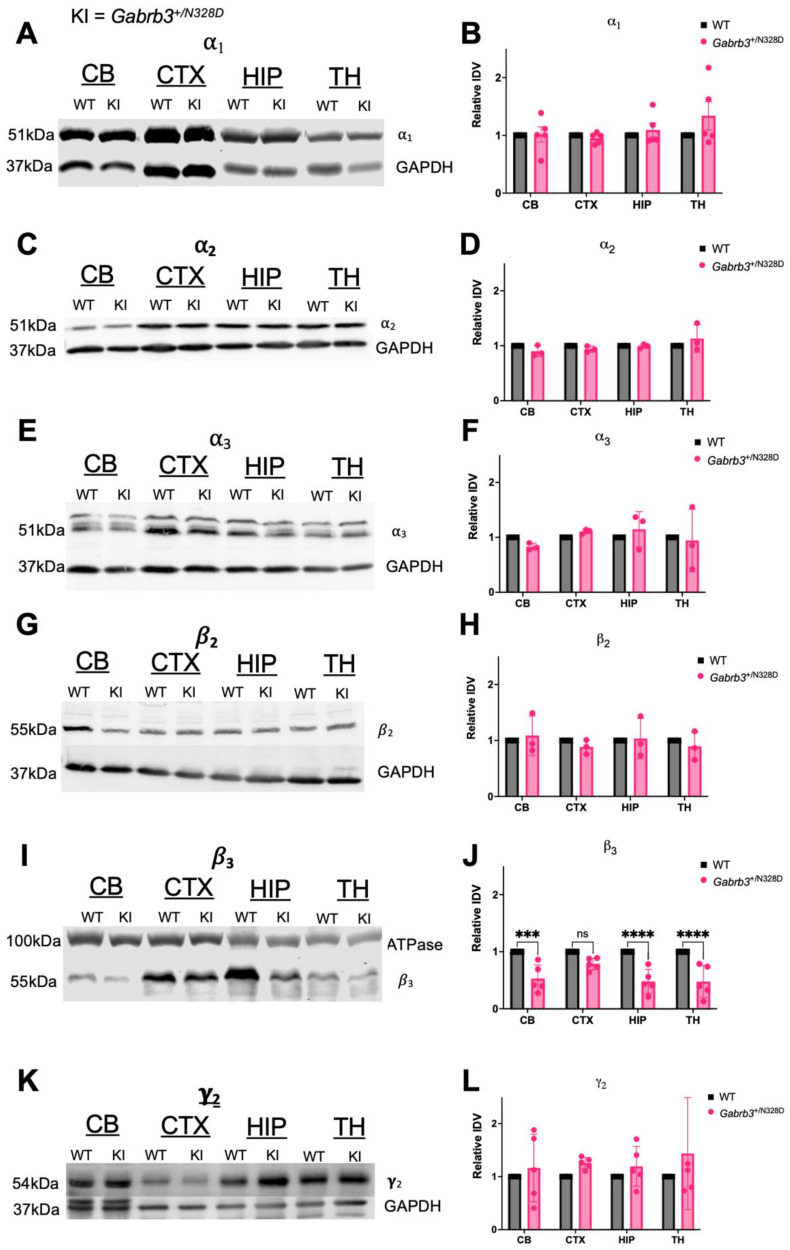
*Gabrb3^+/N328D^* mice show reduction in GABA_A_R β_3_ subunit expression. (**A**,**B**) α_1_ (Ceb: 1.014 ± 0.133; Cor: 0.953 ± 0.041; Hipp: 1.094 ± 0.120; Thal: 1.341 ± 0.245, *N* = 5), (**C**,**D**) α_2_ (Ceb: 0.901 ± 0.057; Cor:.940 ± 0.032; Hipp: 0.992 ± 0.022; Thal: 1.132 ± 0.145, *N* = 3), (**E**,**F**) α_3_ (Ceb: 0.830 ± 0.036; Cor: 1.106 ± 0.187; Hipp: 1.145 ± 0.187; Thal: 0.943 ± 0.329, *N* = 3), (**G**,**H**) β_2_ (Ceb: 1.418 ± 0.536; Cor: 0.885 ± 0.073; Hipp: 1.035 ± 0.208; Thal: 0.895 ± 0.148, *N* = 3), (**I**,**J**) β_3_ (*** Ceb: 0.528 ± 0.108; Cor 0.784 ± 0.049; **** Hipp: 0.479 ± 0.094; **** Thal: 0.477 ± 0.125, *N* = 5), or (**K**,**L**) γ_2_ (Ceb: 1.163 ± 0.285; Cor: 1.257 ± 0.048; Hipp: 1.195 ± 0.168; Thal:1.437 ± 0.472, *N* = 5). *** *p* ≤ 0.01, and **** *p* < 0.0001 by two-way ANOVA with post hoc Sidak’s multiple comparisons test. ns = not significant.

## Data Availability

The data presented in this study are available in this article and attached Appendix A.

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
