# Peer review of "GABA_A_ Receptor β3 Subunit Mutation N328D Heterozygous Knock-in Mice Have Lennox–Gastaut Syndrome"

_ijms, 2023, doi:10.3390/ijms24098458_

Round 1
Reviewer 3 Report
Nwosu et al, in their manuscript describe the development of a mouse model for Lennox-Gastaut Syndrome (LGS), a severe form of epileptic encephalopathy. It usually affects children and progresses with age, reducing the quality of life. LGS is a complex epileptic disorder in which the patients along with seizures suffer from other complications such as cognitive impairment, developmental delay, intellectual disability, and autistic behaviours are seen in some patients.
The authors of this paper have worked to characterize a mouse model for LGS that was based on a recent mutation studied in human patient. This new mutation is in the β3 subunit of the GABAA receptors and the authors have generated a knock-in mouse (Gabrb3+/N328D), that carries this de novo mutation. This manuscript is a very thorough study on the phenotypic analysis of this mouse model.
This paper is well written, and the different assays used in this paper to characterize the knock-in mouse as a viable model to study LGS are appropriate and well explained. Understanding if the mutation alters the subcellular localization of the GABAA receptor in the brain is important to understand the molecular basis of this disease and the authors have suggested this as their future plan.
The authors also propose a potential use of the mouse model to test therapeutic options for LGS. A minor suggestion in that regard is the analysis of the phenotype of Gabrb3+/N328D at older ages (~6m) to study if the phenotype worsens and how effective the treatment can be at that stage of the disease.
Author Response
We appreciate the feedback that the reviewer has provided. We hope that this mouse model can push forward the delineation of epileptic pathophysiology and provide a model system for future therapeutic testing.
Round 2
Reviewer 1 Report
This manuscript by Nwosu and colleagues characterized a novel knock-in mouse model that carries a human mutation associated Lennox-Gastaut syndrome/Developmental Epileptic Encephalopathy. The data presented are generally of good quality. The revision has been improved. However, there are still some issues:
1. Title: beta3 was deleted. Without it, the title became vague and incomprehensible. So it needs to be restored.
2. Abstract: Gabrb3 needs to be defined at the first appearance.
3. The loss of consciousness is difficult to define in mice. Without sleep cycle system, the authors are making subjective judgement and investigator bias is unavoidable. So these statements should be completed removed.
4. Results: the heading for the second section should be 2.2, not 22.2.
5. Fig.5K, the labels were misaligned with the gel lanes.
6. The statement regarding regional specific reduction of beta3 expression appears to be too strong. Although the change in cortex is not statistically significant, there is a clear trend of reduction of beta3 expression.
Reviewer 2 Report
The authors have successfully addressed all of the reviewer's comments.
